# Groin Hernia Repair during the COVID-19 Pandemic—A Romanian Nationwide Analysis

**DOI:** 10.3390/medicina59050970

**Published:** 2023-05-17

**Authors:** Nicolae Dragos Garofil, Mircea Nicolae Bratucu, Mihai Zurzu, Vlad Paic, Anca Tigora, Virgiliu Prunoiu, Alexandru Rogobete, Ana Balan, Cristian Vladescu, Victor Dan Eugen Strambu, Petru Adrian Radu

**Affiliations:** 1Faculty of Medicine, “Carol Davila” University of Medicine and Pharmacy, 050474 Bucharest, Romania; dragosgarofil@gmail.com (N.D.G.); bratucu_mircea@yahoo.com (M.N.B.); zurzu_mihai@yahoo.com (M.Z.); vlad_paic@yahoo.com (V.P.); virgilprunoiu@yahoo.com (V.P.); ana.balan@stud.umfcd.ro (A.B.); drstrambu@gmail.com (V.D.E.S.); drradupetru@yahoo.com (P.A.R.); 2Faculty of Medicine, “Victor Babeș” University of Medicine and Pharmacy, 300041 Timișoara, Romania; alexandru.rogobete@umft.ro; 3National Institute of Health Services Management, 030167 Bucharest, Romania; cristian.vladescu@gmail.com

**Keywords:** groin hernia, elective surgery, COVID-19 pandemic, Romania

## Abstract

*Background and Objectives*: Groin hernia repair surgery (GHRS) is among the most common elective interventions. The aim of this three-year nationwide study on GHRS is to provide a thorough analysis of the impact that the COVID-19 pandemic had on the Romanian Health System in regard to elective procedures. *Materials and Methods*: 46,795 groin hernia cases obtained between 2019 and 2021 from the DRG database using ICD-10 diagnostic codes. The data were collected from all 261 GHRS performing hospitals nationwide, including 227 public hospitals (PbH) and 34 private hospitals (PvH). The 42 variables taken into account were processed using Microsoft Excel 2021, applying Chi square, F-Test Two-Sample for variances, and Two Sample *t*-Test. The significance threshold considered was *p* < 0.001. *Results*: Of the grand total of cases, 96.2% were inguinal hernias, 86.8% were performed on men, 15.2% were laparoscopic procedures, and 6.88% were in PvH. Overall, due to the pandemic, the total number of GHRS decreased with 44.45% in 2020 and with 29.72% in 2021 compared to pre-pandemic year 2019. April 2020 shows the steepest decrease in GHRS (91 procedures nationwide). In the private sector, there was an opposite trend with increases in the number of cases by 12.21% and a 70.22% in both pandemic years. The mean admission period (MAP) for all procedures was 5.5 days. There was a significant difference between PbH and PvH (5.75 vs. 2.8 days, *p* < 0.0001). During the pandemic, the MAP in PbH decreased (6.02 in 2019, 5.82 in 2020 and 5.3 in 2021), remaining stable for PvH (2.9 days in 2019, 2.85 days in 2020 and 2.74 days in 2021). *Conclusions*: The COVID-19 pandemic significantly reduced the overall number of GHRS performed in Romania in 2020 and 2021, compared to 2019. However, the private sector thrived with an actual increase in the number of cases. There was a significant lower MAP in the PvH compared to PbH throughout the three-year period.

## 1. Background

With over 150,000 COVID-19 cases per million inhabitants, Romania ranks in the top 50 most affected countries in the world and top 20 in Europe; the most affected nations worldwide being USA, India, Brazil, France, and Germany [1]. In Romania, the pandemic’s first wave led to the imposition of a lockdown starting March 2020 through may, while the second wave lasted from October to December of the same year. In 2021, another surge of cases from March to May represented the third wave, while the fourth one persisted from October to December.

With more than 3 million cases nationwide, in Romania, the SARS-CoV-2 pandemic has significantly affected the number of admissions as well as the amount of medical and surgical activity, especially elective surgery. In 2020, it was estimated that over 28 million elective surgeries would be cancelled worldwide [2]. Although cancer surgery should not have been affected by the COVID-19 safety protocols, a statewide study conducted in Illinois found that breast cancer surgery decreased by 6.1% and colorectal cancer surgery decreased by 11.6% in 2020 compared to 2019, with the most affected months being April 2020 (−48.3% colorectal cancer surgery) and May 2020 (−45.2% breast cancer surgery) [3].

Groin hernia (GH) is one of the most common general surgery procedures, with over 800,000 of them performed annually in the United States and over 20 million in the Western world [4]. In Romania, GH incidence was between 140 and 240 per 100,000 inhabitants in 2018 [5].

Groin hernia refers to both inguinal and femoral hernia, one of the most common surgical conditions, considering that almost a third of men (27%) will experience it in their lifetimes [6]. Inguinal hernias are nine times more frequent in males than in females, while femoral ones are mostly seen in women, although they only account for 3% of groin hernias [7].

Knowing that groin hernia repair surgery (GHRS) is a common and widespread procedure [8,9] and in over 90% of cases an elective procedure [6], the management of these cases during pandemic times gives an enlightening insight on how elective procedures were handled by the Romanian healthcare system in a moment of crisis.

In Romania, hospital admissions in both public and private hospitals are financed by the National Health Insurance House through the DRG (Diagnosis-Related Group), a classification system based on the complexity and the nature of the medical service provided for different types of patients [10]. The DRG system is administered by the National Institute of Health Services Management (NIHSM). During the beginning of the pandemic, the crisis was managed by central authorities, three months later becoming decentralized [11].

As the pandemic completely shifted the healthcare system’s priorities, the number of elective surgeries performed in the private sector rose in an effort to alleviate the burden on the public hospitals. The aim of this three-year, nationwide study of groin hernia repair during 2019–2021 is to evaluate the COVID-19 pandemic’s impact on elective surgery in our country since hernia groin repair is mainly an elective procedure in Romania.

## 2. Materials and Methods

The nationwide retrospective database study evaluated 46,795 groin hernia repair cases for 45,614 unique patients, obtained between 2019 and 2021 from the anonymized DRG database, provided by the NIHSM. The data were collected from all 261 hospitals in which GHRS is performed, including 227 public hospitals (PbH) and 34 private hospitals (PvH), 64 of them being academic hospitals.

Of all the continuous hospitalizations in the three year period, the following hernia groin procedures records were extracted using ICD-10 diagnosis codes: laparoscopic cure of unilateral femoral hernia (J12701), laparoscopic cure of bilateral femoral hernia (J12702), surgical cure of unilateral femoral hernia (J12703), surgical cure of bilateral femoral hernia (J12704), laparoscopic cure of unilateral inguinal hernia (J12601), laparoscopic cure of bilateral inguinal hernia (J12602), surgical cure of unilateral inguinal hernia (J12603), and surgical cure of bilateral inguinal hernia (J12604).

The study population’s characteristics comprise age, gender, comorbidities, including COVID-19. Also available in the database was the type of hospital, public and private, the name of primary and secondary diagnostics and procedure codes, as well as the admission and discharge dates.

The 42 variables taken into account were processed using Microsoft Excel 2021, applying Chi square, F-test two-sample for variances and *t*-Test. Two-Sample *t*-Test assuming unequal variances after logarithmic transformation of skewed data was applied using the same software. Our study, which is a quantitative exploratory study without a predetermined hypothesis, utilized a *p*-value threshold of <0.001 to minimize the risk of false positives.

The study was approved by the Carol Davila University of Medicine and Pharmacy ethics committee (107/8, 13 April 2023.), which evaluated the study’s methods and confirmed that the use of an anonymized surgical procedures database was appropriate and that obtaining informed consent was not necessary.

## 3. Results

### 3.1. Descriptive Analysis

A detailed count of the interventions, age, gender, procedure, hospital, and death is shown in Table 1. Of the interventions, inguinal hernias (DRG code K40) account for 96.2%, femoral hernias (DRG code K41) for 3.8%. The majority of GHRS were performed on men (86.8%). Female patients accounted for 10.69% of inguinal hernia repairs and 69.81% of femoral hernia repairs.

Of all procedures performed 84.8% had a classic approach and 15.2% were laparoscopic. Bilateral hernia diagnosis accounted for 7.7% of procedures, while 63.3% of them were simultaneous bilateral hernia repairs. Almost a third (29.6%) of all hernias were irreducible/incarcerated hernias and 0.9% were strangulated. Of the total number of procedures, 6.89% were performed in private hospitals.

The incidence of diagnosed irreducible/incarcerated hernia showed a notable rise with advancing age of patients, starting from 15.7% in the 0–9 age bracket and reaching 28% in the 20–60 age range. Subsequently, the incidence increased further to 33% in the 70–80 age range, 44.9% in the 80–90 age range, and peaked at 57.5% in the 90+ age category.

The occurrence of strangulation and gangrene was much higher in cases of femoral hernia, with 7.8% of femoral hernias showing signs of strangulation, in comparison to only 0.013% of inguinal hernias.

The age of the patients is represented in Figure 1. Higher incidence is observed in the 0–9-year age group and in the 60–69-year-old adult individuals. Female patients accounted for 10.69% of inguinal hernia repairs and 69.81% of femoral hernia repairs.

As for the number of admissions per patient, 44,472 of them were admitted once, 1115 were admitted twice, 25 were admitted three times and two patients had four admissions. The mean admission period was 5.55 days, the preoperative care spanned 2.23 days, and the postoperative period was 3.32 days.

The most frequent recorded secondary diagnoses were prostate adenoma (715 patients), obesity (580 patients), diabetes (577 patients), congestive heart failure (501 patients), COVID-19 (531 patients), and atrial fibrillation (504 patients). The number of deaths recorded in total was 162.

### 3.2. The Evolution of Cases during the Pandemic—Total Cases

GHRS cases totaled 46,795 during the three-year period, 20,722 in 2019, 11,510 in 2020, and 14,563 in 2021, respectively. The pandemic’s influence on the number of GHRS is shown in Figure 2 and Figure 3.

Overall, due to the pandemic, the total number of GHRS decreased by 44.45% in 2020. Although 2021, bringing a significant improvement. At the end of the year, there was still a 29.72% decrease compared to 2019.

Regarding complicated hernias evolution, we found that irreducible/incarcerated hernias decreased by 36% in 2020 and by 18.4% in 2021, while strangulated hernias decreased by 47% in 2020 and by 18.9% in 2021.

In the first months of 2020, before the imposition of a lockdown in Romania, January and February registered an increase in interventions compared to the same months of 2019 (103% and 106% respectively). The first wave caused the steepest decrease of surgical procedures. April 2020 was the most affected month, counting only 6% of procedures performed in April 2019 (98 vs. 1655). In July of the same year, the number of surgeries rose to 1146, accounting for 66% of 2019’s month of July.

In the second wave, December 2020 registered a decline to 508 surgeries (33% of December 2019), although by January 2021 the number almost doubled (972) while still representing only half (51%) compared to January 2019.

During the third wave, the number of procedures remained rather stable (970 in May 2021, 59% of May 2019). The third quarter of 2021 recognizes a significant increase in surgeries (1830 in July 2021), outperforming by 10% the month of July 2019 (1726), only to be followed by another steep fall in cases in the pandemic’s fourth wave.

October counted the lowest number of procedures in 2021 (711 and 43% of October 2019). Similar numbers are valid for November (741 and 37% of November 2019).

In comparison to the adult population, pediatric patients (age 0–9) undergoing GHRS showed the same trend in 2020, but with a slightly better recovery in 2021, with a decrease of only 21.8% of cases compared to 2019.

### 3.3. The Evolution of Cases during the Pandemic—Public vs. Private Hospitals

A side-by-side comparison of the number of cases in private hospitals and public hospitals presented in Figure 4 and Figure 5 reveals opposite trends during the pandemic. In the public hospitals, there was a significant decrease in the number of interventions throughout the pandemic, except for the third quarter of 2021, which was similar to the 2019 baseline.

In the private sector, except for the first three months, in which there was a drop in the number of performed procedures, for the rest of the pandemic, there was a significant increase. Overall, the private sector obtained a 12.21% increase in 2020 and a 70.22% increase in 2021, compared to 2019.

### 3.4. Mean Admission Period—PbH vs. PvH

The minimum and maximum lengths of stay for public hospitals were one and 75 days respectively, while for private hospitals they were one and 57 days. The mean admission period (MAP) for all the described procedures was 5.5 days. A significant difference (*p* < 0.0001) lies in the type of hospital the procedure was performed in, e.g., MAP in PbH was 5.75 days and 2.8 in PvH. During the three year-period, MAP remained stable in private hospitals (2.9 days in 2019, 2.85 days in 2020 and 2.74 days in 2021) and decreased in public hospitals (6.02 in 2019, 5.82 in 2020, and 5.3 in 2021).

The difference between public and private hospitals consists in the number of postoperative admission days. In PbH, the mean postoperative period was 3.66 days in 2019, 3.45 in 2020, and 3.10 in 2021. Although it decreased by half a day during the pandemic, it is still averagely two days longer than the one in PvH, which was 1.79 in 2019, 1.67 in 2020, and 1.58 in 2021.

Another explanation for the MAP difference between the two types of hospitals is the preoperative length of stay, which in the PvH is under 1.2 days (1.1 in 2019, 1.17 in 2020, and 1.15 in 2021), while in PbH It averaged on 2.3 days (2.36 in 2019, 2.36 in 2020, and 2.18 in 2021).

### 3.5. Classic vs. Laparoscopic

Of the 46,795 groin hernia repairs, 39,684 (84.8%) were classic interventions and 7111 (15.2%) were laparoscopic. Of the latter, 22.9% were performed in PvH. Thus, in the private sector, 50.59% of GHRS are laparoscopic, while in the public sector, laparoscopic procedures account for only 12.58%. The detailed count of classic or laparoscopic approach with respect to the type of hospital is shown in Table 2.

As shown in Table 3, the number of laparoscopic hernia repairs decreased in 2020 (1842) compared to 2019 (2757), but in 2021 (2512) it almost equaled pre-pandemic times. For the classical approach, the number of procedures lowered in 2020 (9668 vs. 17,965 in 2019) and only partially reached the pre-pandemic count (12,051). However, the percentage of laparoscopic procedures out of the total number of GHRS increased from 13.3% in 2019, to 16% in 2020, and 17.2% in 2021.

The utilization of laparoscopic groin hernia repair surgery in pediatric patients (age 0–9) is relatively low compared to the adult population, with only 7.7% of the total procedures being laparoscopic, and a marginal increase of 1% in 2021 compared to 2019. This low utilization can be attributed to the fact that only nine laparoscopies were performed in private hospitals over the three-year period.

### 3.6. Geographic Distribution

As for the geographic distribution of the surgical hernia repairs, we notice their concentrations in the large academic cities, e.g., Bucharest, Cluj, Iasi, Mures, Timisoara and Craiova, as seen in Figure 6. In 2019, 9379 of the 20,722 procedures (45%) were performed in the six university centers. In 2021, this percentage rose to 47%, without recording statistically significant differences between the pre-pandemic period and pandemic years. Furthermore, it is noticeable that in some districts, such as Giurgiu, Botosani, Calarasi, Ialomita, and Tulcea, there are less than 100 surgeries carried out annually.

### 3.7. Comorbidities and Mortality

Death occurred in seven cases in the laparoscopic group and in 155 cases in the classical group. Laparoscopic hernia repair was associated with a death rate of 0.1%, much lower than the 0.39% of the classical approach (*p* = 0.000112). Female patient mortality (1%) was significantly higher (*p* < 0.001) than that of males (0.24%).

Of the recorded comorbidities, only three were relevant to mortality. Death was more frequently observed in atrial fibrillation patients (*p* < 0.001), chronic kidney disease patients (*p* < 0.001), and in those over 70 (*p* < 0.001). As for risk factor association, patients with atrial fibrillation over the age of 70 (5% of atrial fibrillation patients) had the highest mortality (*p* < 0.001).

During the pandemic (March 2020–December 2021) 531 out of the 26,064 patients tested positive for SARS-CoV-2 during hospital admission. In the same period, there were 94 deaths, five of which were COVID-19 patients. The mortality rate amongst COVID-19 patients was significantly higher (0.94%) than non-COVID patients (0.34%, *p* = 0.024).

## 4. Discussion

The present study is the first nationwide systematic analysis of groin hernia in Romania, aiming to pave the way for future similar studies. The focus on groin hernia repair surgery, one of the most common elective surgical procedures [3], allows to extrapolate for surgical practices in general.

The COVID-19 pandemic had a significant impact on the number of groin hernia repairs performed in Romania. The results of this study showed a decrease in the number of hernia surgeries performed in 2020 compared to 2019, with a partial recovery observed in 2021. The fact that the reduction in groin hernia repair surgeries was less significant during the more severe third and fourth waves of the pandemic compared to the first wave suggests that the decision to completely halt elective surgeries during the lockdown months of March and April 2019 was a questionable one. Although we noted a relatively improved recovery for groin hernia repair surgery (GHRS) in pediatric patients in 2021, the pandemic had a severe impact on this age group, which was considered low risk for COVID-19. The implementation of various measures in hospitals during the pandemic, such as establishing clear criteria for essential procedures, conducting preoperative COVID-19 testing, enhancing infection control measures, improving communication with patients and families, prioritizing procedures based on urgency and need, and ensuring adequate supplies of personal protective equipment, can facilitate the safe continuation of elective surgeries despite the impact of such crises, ultimately benefiting patients. [12]

The impact of the drop in elective procedures during the pandemic can be significant, both in the short-term and long-term. In the short-term, patients who were waiting for elective procedures may have experienced delayed care, which can result in a worsening of their condition or symptoms. In the long-term, there could be a backlog of elective procedures that need to be performed, which may result in longer wait times for patients.

The reduction in elective surgical procedures observed in Romania during the COVID-19 pandemic is a global phenomenon that has been observed in numerous other countries. Overall, during the most disruptive 12 weeks of the pandemic, the estimated reduction due to postponement or cancellation of elective surgery was between 72.3% and 74.4% worldwide [13,14]. A Finnish registry study reported an overall 8% increase in waiting time for elective procedures in 2020 compared to years 2017–2019, although the monthly waiting time increased by 7% to 34% for the months of May–November [15].

For procedures, such as gynecological cancers, although the mean admission period lowered as well, the number of procedures didn’t decrease as steeply as in Romania. In a Dutch study, the number of cervical cancer surgeries decreased by 49% during 2020′s most intense COVID-19 wave compared to 2018–2019 previous data [16].

A study of elective interventions in Maryland during the pandemic reported a 55.8% decrease in procedures [17]. Similar reduction (64.8%) in surgery volume during the first wave was found in an Italian thyroid surgery study [18]. Data from private hospitals in Australia addressing urological elective procedures reveal a 22.6% reduction during April 2020, compared to 2019 [19]. In contrast, the number of urological procedures in Brazil lowered by 86.9% [20].

A 91% decrease in elective surgeries was reported in a study from the United States, allowing for 78% of ICU beds to be available for COVID-19 patients [21]. Another American study found that 71.8% of live donor kidney transplants were postponed, the same being true for two thirds of liver transplantation [22].

It is estimated that in order to get back to date with the surgeries postponed or cancelled during the 12 weeks peak of COVID-19 disruption, it would take 45 weeks post-pandemic if surgery volume increases by 20% compared to its baseline [23]. The Finnish registry study found that elective surgery incidence was 22% higher during the second half of 2020, after the mid-wave decrease and the recovery during May and June [15].

The economic burden will weigh heavily, especially on low-middle income countries, which could lead to a deepening of the already existing disparities in healthcare. Multiple measures and guidelines are needed for preventing further transmission of SARS-CoV-2 and avoiding future spikes in cases, for safely resuming elective surgery, and for planning a sustainable post-pandemic resolution of postponed and cancelled surgeries [24].

The study showed that during the pandemic, private hospitals in Romania, which were not significantly involved in treating COVID-19 patients, actually thrived and performed better, obtaining a surgery count increase of 12.21% in 2020 and 70.22% in 2021, mainly due to the reduced competition from public hospitals. However, it is worth noting that they only account for 6.89% of all GHRS.

The study revealed that patients who underwent groin hernia repair in private hospitals had a significantly shorter mean length of stay compared to those treated in public hospitals, regardless of the epidemiological situation. This finding could be attributed to various factors, such as differences in patient selection, surgical technique (laparoscopic procedures account for 50% of total GHRS in private hospitals), and postoperative care protocols. In contrast, public hospitals in Romania may have a larger patient population with more complex medical conditions or lower socioeconomic status, which could contribute to longer hospital stays. Additionally, the DRG-based financing system of public hospitals fails in stimulating the lowering of the length of stay.

Laparoscopic procedures increased from 13.3% in 2019, to 16% in 2020 and 17.2%, in 2021, in spite of initial concern of a higher risk of infection, due to the aerosol generating nature of the intervention. A systematic review soon demonstrated the lack of evidence behind the suspected risk [11], although the use of personal protective equipment for the entire OR staff was highly recommended [12].

The growing number of laparoscopic procedures in the study is a positive trend for Romania, as it offers patients a less invasive and potentially more effective option for groin hernia repair. Laparoscopic repair has been shown to have lower rates of postoperative pain, shorter hospital stays, faster recovery times, and thus to be less expensive than open-mesh surgery [25,26,27]. A 2015 Romanian study found laparoscopic GHRS to have shorter postoperative care periods by 1.8 days [28]. Patients undergoing laparoscopic groin hernia repair report more satisfaction in comparison to those undergoing the open procedure [29]. Lowering the hospital stay, rate of complications, cost, and quality of care could be improved in public hospitals by performing more laparoscopic GHRS. However, the scarcity of laparoscopic procedures in pediatric patients, particularly in the private sector, found in our study is a cause for concern.

The nationwide study showed an important clustering of patients in university centers and a low patient volume in other hospitals across Romania. While it is true that specialized centers with high case volume have been shown to have better surgical outcomes for hernia repair [30,31,32], it remains crucial to evaluate the regional demand for surgical training facilities and surgeons to promote a more balanced allocation of healthcare resources. Such an approach would ensure better accessibility to healthcare for Romanian patients and improve the overall outcomes.

One of the difficulties in performing the present nationwide analysis was the lack of a national hernia registry. As of 2018, the only such national registry which succeeded in systematically reporting all hernia cases was the Danish Hernia Registry. Similar national initiatives are the Swedish Hernia Registry, EVEREG (Spain), and Club Hernie (France), AHSQC (USA), while other projects are international registries, such as Herniamed (for German speaking countries), EuraHS (Europe). Except for the Danish Hernia Registry where patient participation is compulsory, all other initiatives are voluntary, implying that the responsibility of systematic data entry relies solely on the participating surgeons and hospitals [33]. South Africa also recently started a hernia registry project, HIG (SA), although the one-year follow-up was modest [34].

While the use of the DRG database presents a valuable opportunity for analyzing healthcare utilization and trends in Romania, it is important to acknowledge the limitations of using administrative data for research purposes. One major limitation is the ICD-10 coding system’s inability to capture all clinical details of a disease, including important clinical information, such as hernia severity, repair methods, and postoperative complications. As a result, it is challenging to draw conclusions regarding the effectiveness of different hernia repair procedures. Another limitation of using DRG databases is the potential for overcoding, which can occur when medical professionals artificially increase reimbursement rates. This may have led to the pre-pandemic overdiagnosis of irreducible/ incarcerated and strangulated hernias. In our study, the percentages of complicated hernias are significantly higher than what is reported in the literature [6]. Surprisingly, we found that incarcerated and strangulated hernias followed the same overall descending trend during the pandemic, contrary to what was expected. This issue was reported also by Anne S Wing et al. [35] and Lima et al. [36], raising the question of the possible overuse of emergency hernia surgery. The retrospective design of the study also limits the ability to control for confounding variables and establish causality. Lastly, the study was only able to report short-term outcomes, such as 30-day readmission and mortality rates, and could not provide data on long-term outcomes, such as recurrence rates, morbidity, chronic pain, or quality of life.

The World Health Organization has developed ICD-11, which is expected to address some of the limitations of ICD-10 and provide a more accurate and comprehensive coding system [37]. Implementing ICD-11 worldwide, including in Romania, could improve the accuracy of future studies and help better understand the epidemiology of various diseases and procedures. To fully understand the burden of groin hernia repair and its outcomes in Romania, the establishment of disease registries that capture patient-level data is crucial.

## 5. Conclusions

There was a significant decrease in the total number of hernia repairs performed during the pandemic period in Romania, which suggests a reduction in elective surgery overall.

Private hospitals saw a rise in the number of cases during the pandemic, and they also had a significantly shorter mean admission period for hernia repairs compared to public hospitals, suggesting potential differences in resource allocation and patient management between the two sectors.

The proportion of laparoscopic hernia repairs increased during the pandemic period, indicating a shift towards less invasive surgical techniques.

Despite the limitations of the DRG database, it can still provide valuable insights into trends and patterns in surgical procedures, particularly in countries where disease registries are not available.

More research is needed to fully understand the long-term implications of the COVID-19 pandemic on elective surgical procedures, as well as the potential impact on patient outcomes and health system resources.

## Figures and Tables

**Figure 1 medicina-59-00970-f001:**
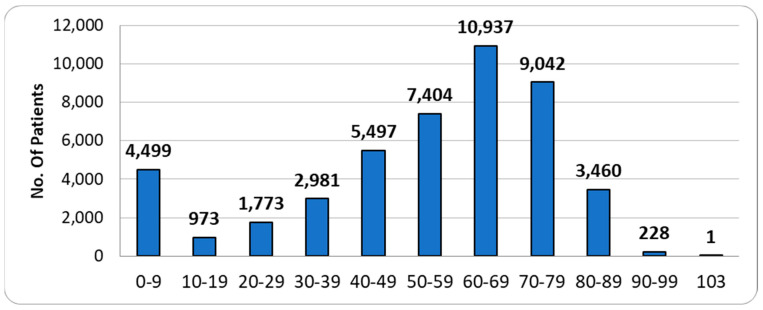
Groin hernia repairs 2019-2021—patients age groups.

**Figure 2 medicina-59-00970-f002:**
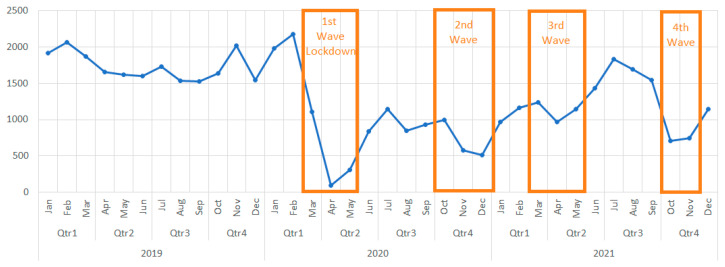
Evolution of the total number of cases in 2019, 2020, 2021.

**Figure 3 medicina-59-00970-f003:**
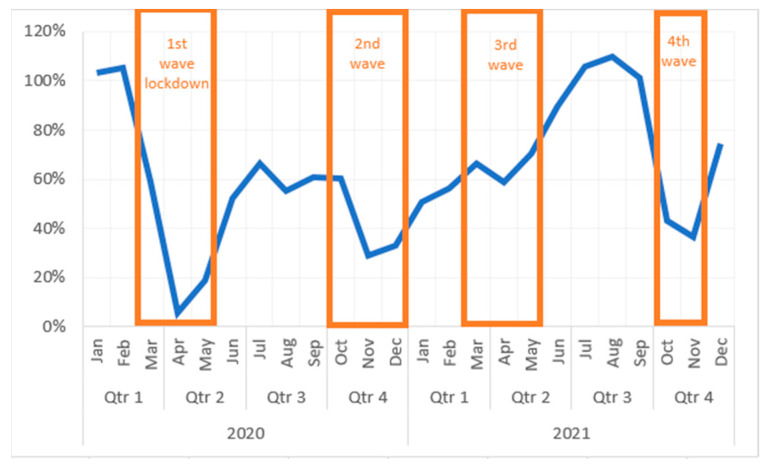
Evolution of the number of cases (%) relative to 2019.

**Figure 4 medicina-59-00970-f004:**
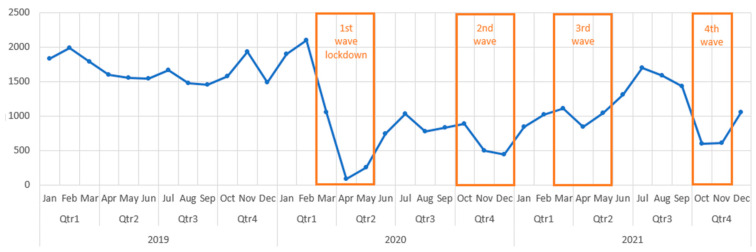
Number of cases in public hospitals in 2019, 2020 and 2021.

**Figure 5 medicina-59-00970-f005:**
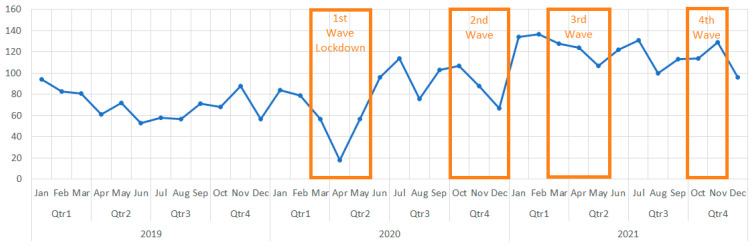
Number of cases in private hospitals in 2019,2020 and 2021.

**Figure 6 medicina-59-00970-f006:**
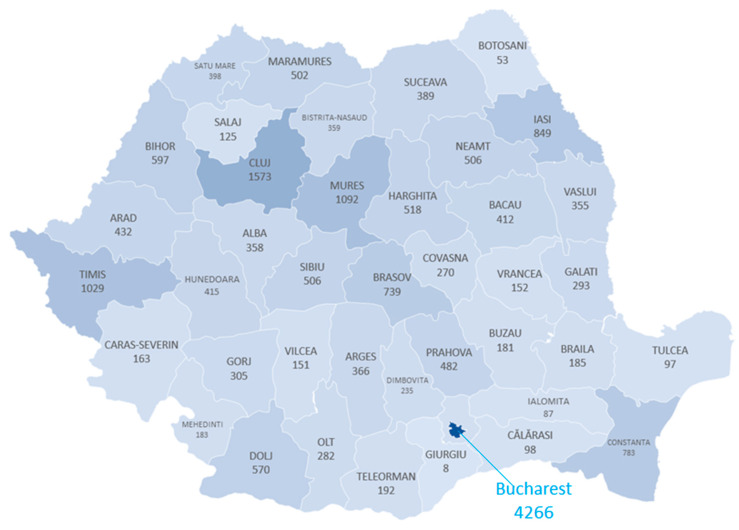
Number of cases of Groin hernia repair in the counties of Romania in 2019.

**Table 1 medicina-59-00970-t001:** Descriptive Analysis.

		2019	2020	2021	Grand Total
**No. of Surgical Interventions**		**20,722**	**11,510**	**14,563**	**46,795**
	Inguinal Hernia	19,955 (96.3)	11,055 (96)	14,010 (96.2)	45,020 (96.2)
	Femoral Hernia	767 (3.7)	455 (4)	553 (3.8)	1775 (3.8)
	Bilateral Hernia Diagnoses	1457 (7)	971 (8.4)	1176 (8.1)	3604 (7.7)
	Simultaneous Bilateral Hernia Repair	903 (4.4)	606 (5.3)	775 (5.3)	2284 (4.9)
**Complications**					
	Irreducible/ Incarcerated	5662 (27.3)	3590 (31.2)	4619 (31.7)	13,871 (29.6)
	Strangulated	185 (0.893)	98 (0.851)	150 (1.03)	433 (0.925)
**Age**					
	0–9	1926 (9.3)	1068 (9.3)	1505 (10.3)	4499 (9.6)
	10–19	457 (2.2)	208 (1.8)	308 (2.1)	973 (2.1)
	20–29	779 (3.8)	441 (3.8)	553 (3.8)	1773 (3.8)
	30–39	1304 (6.3)	760 (6.6)	917 (6.3)	2981 (6.4)
	40–49	2391 (11.5)	1393 (12.1)	1713 (11.8)	5497 (11.7)
	50–59	3162 (15.3)	1873 (16.3)	2369 (16.3)	7404 (15.8)
	60–69	4976 (24)	2692 (23.4)	3269 (22.4)	10,937 (23.4)
	70–79	4036 (19.5)	2188 (19)	2818 (19.4)	9042 (19.3)
	80–89	1578 (7.6)	836 (7.3)	1046 (7.2)	3460 (7.4)
	90–99	112 (0.5)	51 (0.4)	65 (0.4)	228 (0.5)
	103	1			1
**Gender**					
	F	2796 (13.5)	1490 (12.9)	1870 (12.8)	6156 (13.2)
	M	17,926 (86.5)	10,020 (87.1)	12,693 (87.2)	40,639 (86.8)
**Procedure**					
	Classic	17,965 (86.7)	9668 (84)	12,051 (82.8)	39,684 (84.8)
	Classic bilateral	537 (2.6)	283 (2.5)	382 (2.6)	1202 (2.6)
	Laparoscopic	2757 (13.3)	1842 (16)	2512 (17.2)	7111 (15.2)
	Laparoscopic bilateral	366 (1.8)	323 (2.8)	393 (2.7)	1082 (2.3)
	Relapsed hernia	1443 (7.0)	901 (7.8)	1110 (7.6)	3454 (7.4)
**Hospital**					
	Private	843 (4.1)	946 (8.2)	1435 (9.9)	3224 (6.9)
	Public	19,879 (95.9)	10,564 (91.8)	13,128 (90.1)	43,571 (93.1)
**Outcome**					
	Death	59 (0.285)	43 (0.374)	60 (0.412)	162 (0.346)

The percentage of surgical interventions is noted in parenthesis

**Table 2 medicina-59-00970-t002:** Classic vs. laparoscopic—private and public hospital.

	Classic	Laparoscopic	Total	Laparoscopic %
Private	1593	1631	3224	50.59%
Public	38,091	5480	43,571	12.58%
Total	39,684	7111	46,795	15.20%

**Table 3 medicina-59-00970-t003:** Classic vs. laparoscopic—no. of cases by year.

	2019	2020	2021
Classic	17,965	9668	12,051
Laparoscopic	2757	1842	2512
% Laparoscopic	13.30%	16%	17.2%

## Data Availability

Not applicable.

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
