# Peer review of "Groin Hernia Repair during the COVID-19 Pandemic—A Romanian Nationwide Analysis"

_medicina, 2023, doi:10.3390/medicina59050970_

Round 1

Reviewer 1 Report

The Authors present an overview of the GH surgery statistics in Romania during COVID-19 pandemic. 

The introduction provides sufficient data about the procedure and the problem, however I suggest incorporating some data about other types of surgeries - especially oncological - during the pandemic. The Authors wrote:"This common and widespread procedure allows us to extrapolate with a small error margin on the current surgical practices in general." - why do Authors believe this to be the true statment? What do you base this on? Please specify if there are any relevant statistics which could support that sentence.

Regarding the methods section: any specific reason why significance threshold was set on p<0,001 instead of 0,05?

In Table 1. I recommend including % in addition to the numbers

In discussion the Authors claim:"In hindsight, the decision to suspend elective surgeries during the March and April 2020 lockdown months was questionable, and we must learn from this experience to ensure better preparedness for future crises." Could you provide ideas based on your conclusions which if implemented would improve actions during another crisis? What it is that we should learn?

"Surprisingly, incarcerated and strangulated hernias followed the same overall descending trend during the pandemic, contrary to what was expected." - surprisingly indeed, what do you think is the reason?

The Authors correctly indetified limitations of their study. Reported results are in the most part as would be expected, therefore I encourage enriching the paper with some suggestions based on the conclusions from the study.

Author Response

Hello and thank you very much for your review

The manuscript was modified using track changes. All chages are mentioned and explained in the two cover letters adressed to each reviewer. Please see the attachment

Reviewer 2 Report

The present nationwide investigation of the impact of the COVID pandemic on the surgical care of patients (children and adults) with a hernia (inguinal and femoral hernia) in Romania is clearly written. The following main statements are made by the authors Garofil et al. in their retrospective work, which evaluated patient data from 2019 to 2021 provided by the anonymized DRG database.

Symbol „Von der Community überprüft“ In 261 hospitals were 46,795 groin hernia cases operates during the study period. The number fell sharply during 2020 and 2021 when the pandemic was at its peak. However, the number of laparoscopically operated patients in private hospitals increased from year to year.
This is a interesting finding and the authors discussed this issue in detail. Only 2 points need to be addressed. 1. The pediatric inguinal hernia differs greatly from adult patients both in etiology and in the surgical technique. This is not specifically addressed in the work. Why? 2. Some current works e.g. from Scotland (Ewing AS, McFadyen R, Hodge K, Grossart CM, East B, de Beaux AC. The Impact of the COVID-19 Pandemic on Hernia Surgery: The South-East Scotland Experience. Cureus. 2022 Sep 24;14(9): e29532. doi: 10.7759/cureus.29532 and USA are not cited. E.g. Lima DL, Pereira X, Dos Santos DC, Camacho D, Malcher F. Where are the hernias? A paradoxical decrease in emergency hernia surgery during COVID-19 pandemic. Hernia 2020 Oct;24(5):1141-1142. doi: 10.1007/s10029-020-02250-2 Please explain this.
Thank you!

Author Response

Hello and thank you very much for your review

The manuscript was modified using track changes. All chages are mentioned and explained in the two cover letters adressed to each reviewer.

Round 2

Reviewer 1 Report

I thank the authors for addressing my comments. I hope you found them helpful in improving the quality of your work.